# Lebanese cannabis oil extract protected against folic acid-induced kidney fibrosis in rats

Diana Bylan[1]☯, Alia Khalil[2]☯, Wassim Shebaby[1], Christabel Habchy[1], Selim Nasser[2], Wissam H. Faour ![ORCID][2]*, Mohamad Mroueh ![ORCID][1]

**1** Pharmaceutical Sciences Department, School of Pharmacy, Lebanese American University, Byblos, Lebanon, **2** Gilbert and Rose-Marie Chagoury School of Medicine, Lebanese American University, Byblos, Lebanon

☯ These authors contributed equally to this work.

\* wissam.faour@lau.edu.lb

**Data Availability Statement:** All relevant data are within the paper and its Supporting Information files.

## Abstract

### Background

Renal fibrosis is a major manifestation of chronic kidney disease. To date, there are no treatments to reverse kidney fibrosis. Cannabis is an aromatic herb that is widely known for its anti-diabetic and anti-inflammatory properties. The aim of this study is to evaluate the protective effect of Lebanese cannabis oil extract (COE) against folic acid (FA) induced renal injury both in vitro and in vivo.

### Materials and methods

A single dose of 250 mg/kg of Folic acid was administered to induce renal fibrosis in rats. COE was injected at varying doses of 5, 10, and 20 mg/kg. Body weight of rats were monitored and clinical parameters including serum creatinine, urea, and electrolytes were measured. Moreover, pathological examination of the kidney and heart was performed. Conditionally immortalized cultured rat podocytes were exposed to high concentrations of folic acid in the presence or absence of COE. MTS and in vitro scratch assay were used to assess podocyte cells viability and migration respectively. Western blot analysis was used to evaluate the phosphorylation levels of AKT and p38 MAPK.

### Results

Rats that received FA showed a marked increase in serum creatinine when compared to the non-treated control group. COE at doses of 5 and 10 mg/kg significantly decreased serum creatinine induced by FA. Serum sodium was significantly reduced in all the groups receiving COE. Furthermore, COE ameliorated renal and cardiac pathology abnormalities caused by FA in a dose-dependent manner. Cell viability assay revealed that COE reversed cytotoxicity induced by FA in rat podocytes. In vitro scratch assay showed that COE partially restored the migratory capacity of podocytes incubated with FA. Dose-dependent

**Funding:** This project is funded by a grant from the Lebanese American University – President Intramural Research Funds (PIRF). The funders had no role in study design, data collection and analysis, decision to publish, or preparation of the manuscript.

**Competing interests:** The authors have declared that no competing interests exist.

experiments showed that COE (1 and 2µg/ml) induced a significant increase of phospho-(S473)-AKT along with a decrease in phospho (T180 + Y182) P38 levels.

## Conclusion

The current results revealed important protective effect of Lebanese cannabis oil extract against folic acid—induced renal fibrosis in rats.

## Introduction

Kidney disease is considered a major health burden worldwide [1]. The number of kidney disease patients increases with the increased incidence of other chronic illnesses including diabetes and hypertension [2]. In Lebanon, the prevalence of dialysis in Lebanon is among the highest in the world [3]. It is estimated at 777 patients per million population compared to 410 dialysis patients per million population worldwide [4]. Renal fibrosis, characterized by tubulointerstitial fibrosis and glomerulosclerosis, is one of the final and most common manifestations of chronic kidney disease CKD. Moreover, it is the principal process underlying the progression of CKD to end stage kidney disease also known as ESKD [5]. Glomerulosclerosis and tubulointerstitial fibrosis result mainly from the dysregulated wound healing process that is due to the imbalance between the excessive synthesis and the reduction in the breakdown of extracellular matrix [6]. Podocytes are highly differentiated cells that play a major role in maintaining the integrity of the glomerular filtration barrier. In the past two decades, numerous studies have suggested that podocytes dysfunction and loss are causative events in the pathogenesis of a multitude of proteinuric kidney diseases including diabetic nephropathy, focal segmental glomerulosclerosis, lupus nephritis and membranous nephropathy [7].

Among several signaling pathways, P38 MAPK and protein kinase B/AKT have been recognized in transducing stress and pro-survival signals in podocytes respectively. Interestingly, blockade of the p38 pathway significantly reduced podocyte foot process effacement, inhibited proteinuria and fibrosis development and improved kidney functions in patients with FSGS [8, 9]. Moreover, in immortalized mouse cell lines and kidney biopsies, p38 MAPK inhibition prevented in vitro actin reorganization and preserved podocyte survival [10]. On the contrary, AKT activation have been shown to sustain podocyte survival in several forms of glomerular diseases [11, 12], and therefore, genetic or pharmacological inhibition have been associated with podocyte death and glomerulosclerosis [13, 14].

Cannabis is one of the oldest plants in the world that belongs to the Cannabaceae family. The most common cannabis species are *Cannabis sativa*, *Cannabis indica* and *Cannabis ruderalis* [15]. Cannabis possesses anti-emetic, diuretic, anti-inflammatory and anti-epileptic properties [16]. COE has been widely used in Lebanon in the management of certain medical conditions mainly cancer, pain, and diabetes [17]. The modulation of cannabinoids receptors has been suggested in different types of renal disease, nephropathies, and renal fibrosis [18].

Nowadays, and despite all the advances, there is no effective therapy that could prevent or reverse renal fibrosis [19], highlighting the urgent need for both studying the pathological mechanisms and testing the efficacy of a variety of compounds or pharmacological agents to fight kidney disease.

In this regard, Folic acid-induced kidney injury animal models, among others, have been invaluable and provided excellent platforms for disease induction and intervention [20]. FA also known as vitamin B9 is a nutritionally beneficial cofactor involved in cellular proliferation

and growth. However, high doses of FA (up to 250 mg/day body weight) as widely used in the induction of animal kidney disease are very toxic resulting in increased blood urea nitrogen (BUN), creatinine and proteinuria [21].

The underlying biochemical and molecular processes are multifaceted and complex including oxidative stress, apoptosis, tubular necrosis, inflammation and impairment of mitochondrial bioenergetics among others [20].

While FA-induced kidney fibrosis in animals is a well-established in vivo model, the effect of higher concentrations of FA on kidney epithelial cells in vitro has not yet been extensively studied. Therefore, the aim of this study is to investigate the therapeutic effect of Lebanese cannabis oil extract on FA induced kidney injury in vivo and in vitro.

## Materials and methods

### Plant collection and oil extraction

Dried samples of Lebanese cannabis strain were provided through Drug Enforcement Office. Plant extract was prepared as previously described [17]. Briefly, a sample of 10 g of air-dried cannabis flower was extracted with ethanol for 48 h. The extract was filtered and concentrated at 45˚C under reduced pressure to yield 1.17 g of COE, and subject to gas chromatography mass spectrometry for chemical analysis.

### Cell culture

Conditionally immortalized rat glomerular epithelial cells (podocytes) kindly provided by Dr. Assaad Eid (American University of Beirut) was cultured as previously described [22]. Briefly, Podocyte cell cultures propagation was made in RPMI media + 10% Fetal Bovine Serum (FBS) and Penicillin-Streptomycin (100 U/ml penicillin, 100 μg/ml streptomycin). At confluency, cells were detached by trypsinization with 1X Trypsin and centrifuged at 1000 rpm for 5 minutes at 20˚C, then re-suspended in fresh complete media. The resuspended cultures were then transferred into new sterile dishes. Induction of growth arrest and differentiation is induced by incubating semi-confluent podocytes without insulin. Accordingly, cell morphology changed from cobblestone to arborized appearance following few days of culture.

### Cell viability assay

Podocyte cells were seeded in 96 well-plates (8000 cells/ well). The following day, the cells were incubated in serum free medium composed of 0.1% FBS. After overnight starvation, cells were treated with increasing concentration of FA for 24h. In other experiments podocyte cells were treated with 25mM FA in presence or absence of COE (0.5, 1 and 2 μg/ml) for 24h. The viability of the cells was assessed using the Cell Titer 96 AQueous Non-Radioactive Cell Proliferation Assay Kit (Promega, USA) which is a colorimetric method based on a reaction of mitochondrial dehydrogenase with 3-(4,5-dimethylthiazol-2-yl)-5-(3-carboxylmethoxyphenyl)-2-(4-sulfophenyl)-2H-tetrazolium inner salt (MTS) as the reagent. Briefly, 20 μL of reagent solution was added to each well of the 96-well plate. After incubation at 37˚C in humidified 5% $CO_2$ for 1 hour, absorbance was read at 490 nm using Elisa microplate reader. Percentage of cell proliferation was determined using the following formula:

$$\frac{\text{Absorbance test well}}{\text{Absorbance control well}} \times 100$$

## Wound healing assay

Podocyte cells were counted using hemocytometer and plated in 12 well plates at the density 1x104/well. The plates were incubated overnight under growth conditions and allowed for cell recovery and exponential growth. After overnight incubation, the cells were serum starved in 0.1% FBS RPMI media for 24 h.

Mechanical scratch representing wound was created in the near confluent monolayer of cells by gently scraping with sterile 200 μL micropipette tip. The cells were then rinsed with serum free RPMI and treated with 1mM FA in the presence or absence of 1 μg/ml COE. Scratch width was photographed at two time points (0 and 24 h) from three fields of view with the 10x objective using light microscopy. The experiments were performed in triplicate. Using the Image J software, the cell-free wound surface was measured between the wound edges, averaged between the fields of views and triplicates and the percentage of wound closure was calculated according to the following equation: $\frac{A0-At}{A0} \times 100$ where "A0" is the initial wound area, "At" is the wound area after "n" hours of the initial scratch, both in $\mu m^2$.

## Western blot

Western blot analysis was done as previously described [22]. Briefly, podocyte cells were counted using hemocytometer and plated in 6 well plates at the density 25x104/well. The plates were incubated overnight under growth conditions and allowed for cell recovery and exponential growth. After overnight incubation, the cells were serum starved in 0.1% FBS RPMI for 24 h. Following overnight starvation, the cells were treated with 25mM FA in the presence or absence of 1 and 2μg/ml COE for 24h.

A total of 106 cells for each treatment were pelleted and lysed in RIPA lysis buffer supplemented with protease inhibitor cocktail (Roche). Cell lysates were centrifuged at 12000g, 4˚C for 15 min and protein concentration was measured using Bradford assay kit (Thermo Fisher). Proteins were separated by SDS–PAGE in reducing conditions and transferred to nitrocellulose membrane. After blocking with 5% milk, the membrane was incubated overnight at 4˚C with the appropriate primary antibody: anti-phospho-AKT1 (S473), anti-phospho-p38 (phospho T180 + Y182), total anti-(AKT1 + AKT2 + AKT3) antibody and total anti-p38 alpha/beta MAPK antibodies (all antibodies were purchased from Abcam). After overnight incubation, the membranes were washed then incubated with HRP-conjugated secondary antibody for 1 h at room temperature and revealed using the ECL substrate using the ChemidocTM MP Imaging System (Biorad). Monoclonal anti-beta actin antibody was used for protein loading control. Densitometric analysis was performed using ImageJ.

## Animals and experimental approach

Eight weeks old male Sprague Dawley rats were acquired from the animal care facility laboratories at the Lebanese American University. The animals had free access to food and water and were kept at 12 hours light to dark cycle. The animal protocols were approved by the IACUC (Institutional Animal Care & use Committee) of the Lebanese American University (**IRB #:** LAU.ACUC.SOP.MM7.1/November/2021), and all animal experiments were done according to the National Institutes of Health guidelines for animal care and handling.

Rats were randomized into 6 groups: group 1 (untreated) (n = 5), group 2 (F.A control) (n = 5), group 3 (FA + 5mg/kg COE) (n = 7), group 4 (FA + 10 mg/kg COE) (n = 6), group 5 (FA + 20 mg/kg COE as treatment) (n = 7), group 6 (20mg/kg COE + FA as prevention) (n = 7). All injections were administered via the intra-peritoneal route. FA was dissolved in

sodium bicarbonate (300 mM) while cannabis oil was dissolved in a mixture of Ethanol: Tween 80: PBS at a 1:1:18 ratio. Group 6 receiving COE as prevention was administered 20 mg/kg of COE from day 1–5 of the experiment while all the other groups received the corresponding vehicle (Ethanol: Tween 80: PBS at a 1:1:18 ratio). On day 6, all the rats were administered FA at a dose of 250 mg/kg of body weight except for rats in group 1 which received the bicarbonate vehicle. Cannabis oil was administered eight days after FA injection daily for 8 days then every other day for one week thereafter. Groups 3, 4 and 5 received COE at doses of 5, 10, and 20 mg/kg of rat body weight respectively while group 1, 2, and 6 received the empty vehicle. Animals were sacrificed after the last injection as follow: animals were deeply anesthetized with pentobarbital (100 mg/kg), then animal surgery was conducted after the induction of anesthesia. The animals were then euthanized by exsanguination.

## Biochemical assays

Blood samples were collected immediately after the sacrifice of rats. EDTA was added to prevent clotting of the samples in a ratio of 25 μL (0.5 M) for each 1 mL of blood collected. Then, blood samples collected were centrifuged at 3000 rpm for 20 min at 4°C for efficient separation and recovery of plasma. The concentrations of the renal function biological parameters including serum creatinine, urea, and electrolytes were analyzed.

## Histopathology

Organ pathology was done as previously described [23]. Briefly, the kidneys and hearts of all groups were dissected out and weighed. The organs were fixed in 10% neutral buffered formalin. Then, they were taken, dehydrated and embedded in paraffin. Tissue sections from the paraffin-embedded blocks were mounted on glass slides, deparaffinized, rehydrated, stained with Hematoxylin & Eosin and then examined microscopically.

## Statistical analysis

Data analysis was expressed as mean ± standard error of mean (SEM). For two group comparison, unpaired t-test was used. Differences between multiple groups were evaluated using one way ANOVA followed by Bonferroni post-hoc test and two way ANOVA when applicable. The significance level was accepted with a $p$-value $<0.05$ (*), $<0.01$ (**), $0.001$(***) and $<0.0001$(****). Statistical analysis was performed using GraphPad Prism 8.4.

## Results

### COE significantly improved cell viability and reversed the cytotoxic effect of folic acid in cultured podocytes

We first evaluated the effect of FA on podocyte cells viability. As shown in Fig 1, podocyte cells were treated with increasing concentrations of FA (1.5, 10, 15, 20 and 25mM) and their cytotoxic effect was investigated using MTS cell proliferation assay. Accordingly, the results showed a dose-dependent cytotoxic effect of FA with an $IC_{50}$ of 25mM after 24 hours of incubation (Fig 1A). To test the protective effects of COE on podocytes against FA cytotoxicity, cells were treated with 25mM FA in the presence or absence of 0.5, 1 and 2μg/ml COE for 24h. Cell viability assay showed that COE attenuated the cytotoxic effect of FA in a concentration-dependent manner (Fig 1B).

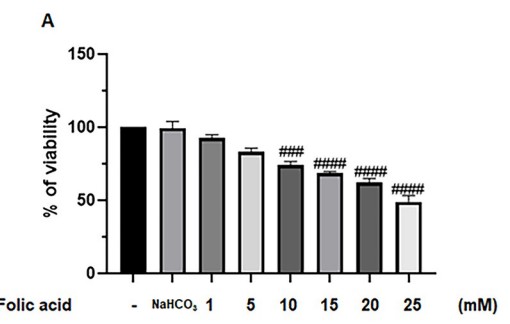
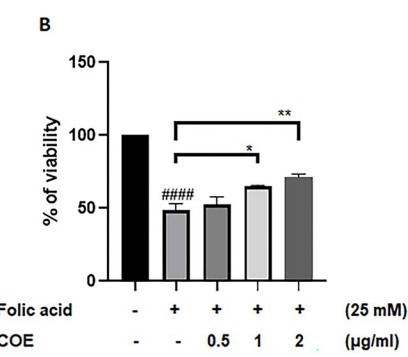

**Fig 1. Cannabis oil extract protected podocyte cells against folic acid induced cytotoxicity.** (A) Rat podocytes were treated with various concentrations of Folic acid alone as indicated, or (B) with folic acid in the absence or presence of Cannabis Oil Extract (COE) at various concentrations for 24 h as indicated. Cell viability was evaluated via Cell Titer 96 Aqueous Non-Radioactive Cell Proliferation Assay. Data are expressed as mean ± SEM (n = 3). Differences between groups were evaluated using one-way ANOVA followed by Bonferroni's multiple comparison test. Significance is accepted when $P<0.05$. ### $P<0.001$, #### $P<0.0001$ vs control, *$P<0.05$, **$P<0.01$ significantly different from the folic acid-only treated group.

## COE partially reversed migration inhibition in podocytes treated with folic acid

In vitro scratch assay was performed in order to measure the rate of podocyte cells migration in the presence of 1mM FA in the presence or absence of 1μg/ml COE. As shown in "Fig 2", COE accelerated wound closure (79.4±3.4%) when compared to control non-treated cells (67.2±2.5%). FA significantly slowed down the closure of the cell free gap after 24 h by 56.4 ± 6.1%. The addition of COE partially reversed the inhibitory effect of FA and restored wound closure by 71.5 ±5% (Fig 2).

## COE reversed folic acid effect on both AKT and P38 phosphorylation levels in podocytes

We further investigated the effect of FA on the phosphorylated activated levels of both AKT and P38. "Fig 3" showed that FA was able to increase the levels of phosphorylated activated form of the proapoptotic P38 MAPK pathway while reduced the levels of the phosphorylated activated form of the pro-survival AKT pathway. COE was able to reverse the effect of FA on both pathways. Thus, COE restored phospho-(S473)-AKT levels inhibited by FA, while it reduced phospho (T180 + Y182) P38 MAPK levels induced by FA in a dose dependent manner.

## Cannabis oil extract significantly modulated serum creatinine, electrolytes and urea levels

FA administered at a dose of 250 mg/kg significantly increased serum creatinine levels when compared to the control group. COE at a dose of 5 mg/kg and 10 mg/kg significantly decreased serum creatinine levels when compared to FA injected animal group. However, the decrease in creatinine level in the rats groups treated with 20mg/kg COE either as treatment or prevention was not statistically significant (Fig 4A). Serum urea levels remained unchanged in the control, FA and COE treated (5 and 10 mg/kg) groups. However, groups receiving 20 mg/kg of COE either as treatment or prevention showed higher serum urea levels when compared to the FA and control groups (Fig 4B). While FA administration had no significant effect on sodium level, animal groups receiving cannabis oil extract at various doses of 5, 10 and 20 mg/kg respectively showed significantly lower serum sodium levels when compared to the FA treated

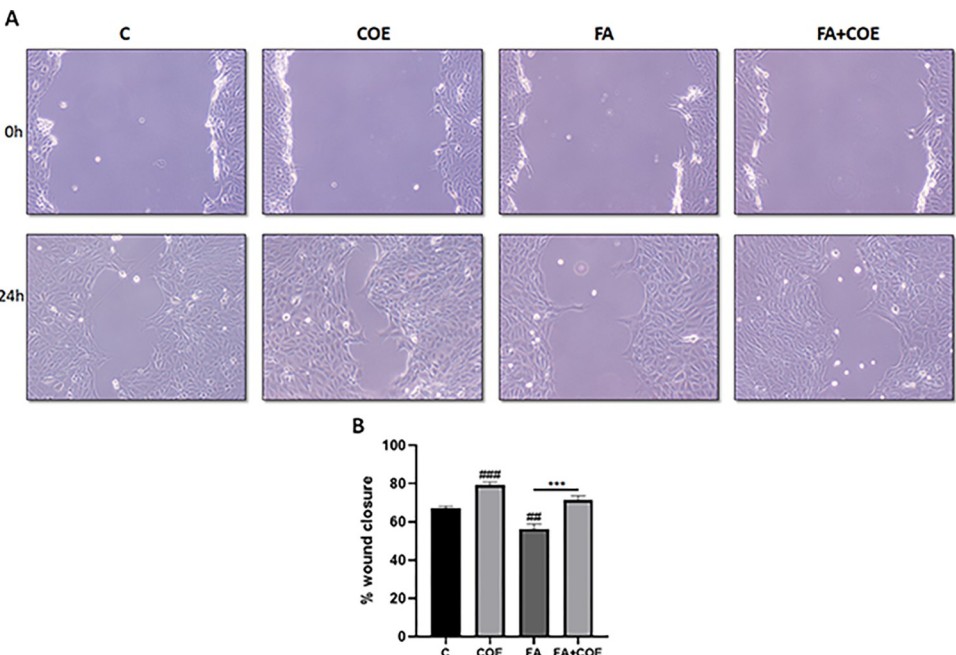

**Fig 2. Podocyte cells migration.** (A) Representative pictures showed the migration of podocyte cells after induction of a scratch representing wound. All the pictures were taken immediately after the scratch was induced (at zero hour) and after 24h of incision. Podocytes on the pictures were cultured in different conditions as previously indicated in material and methods. Pictures were taken at 10X magnification. (B) The relative change in cell-free gap surface was measured at different time points after creation of the wounding scratch and expressed as fold change over zero time. Results are expressed as mean ± SEM of three independent experiments performed in triplicate. Statistical analysis was performed using unpaired t-test. Significance is accepted when $P<0.05$. ### $P<0.001$, ## $P<0.01$ vs control, \*\*\*$P<0.001$ significantly different from the folic acid-only treated group.

group. Finally, levels of serum chloride and potassium showed no significant differences among all groups (Fig 5B and 5C) except for the COE 20mg/kg group in which the potassium level was significantly induced as compared to control group (Fig 5C).

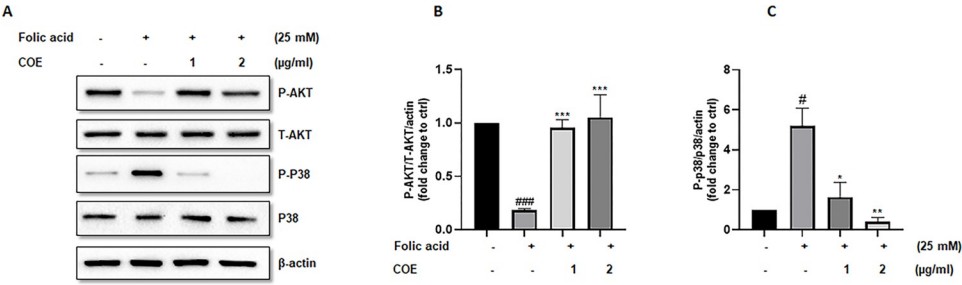

**Fig 3. Folic acid modulation of AKT and p38 MAPK phosphorylation in podocytes.** In (A), cultured immortalized rat podocytes were treated with folic acid in the absence or presence of Cannabis Oil Extract COE (1 and 2μg/ml) for 24hrs as indicated. Cell extracts (50μg) were analyzed by Western blotting using anti-phospho-AKT1 (S473), anti-phospho-p38 (phospho T180 + Y182), Anti-AKT1 + AKT2 + AKT3 antibody and anti-p38-alpha/beta MAPK antibodies. The levels of phospho-AKT-(S473) and phospho-p38 were normalized to total AKT (B) and p38 MAPK (C) respectively then to actin protein content, and the signal intensity was identified by densitometry. Data are expressed as mean ± SEM (n = 3). Differences between groups were evaluated using one-way ANOVA followed by Bonferroni's multiple comparison test. Significance is accepted when $P<0.05$. #$P<0.01$, ### $P<0.001$ vs control, \*\*$P<0.01$, \*\*\*$P < 0.001$ significantly different from the folic acid-only treated group.

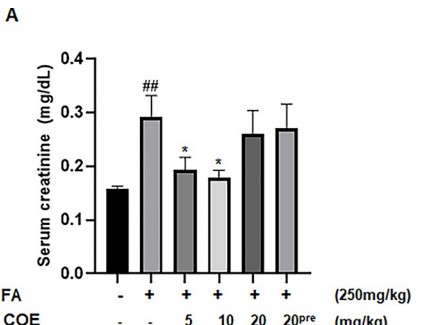
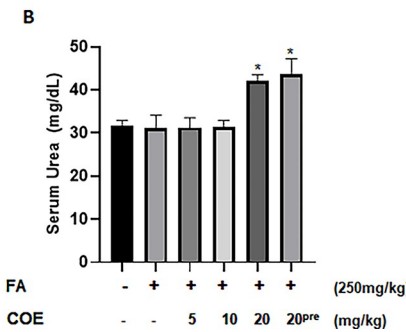

**Fig 4. Average serum creatinine levels (mg/dL) and average serum urea levels (mg/dL) in different rat groups.**
Rats were injected with 250 mg/Kg intraperitoneal folic acid on day 6. COE was administered eight days after the FA injection daily for 8 days then every other day for one week thereafter. In some experiments (COE 20^pre) rats were first injected with 20 mg/kg cannabis oil and at day 6 folic acid is administered. In (A), serum creatinine and in (B), urea levels were measured respectively. Each column represents the mean ± SEM of seven animals. Statistical analysis was performed using unpaired t-test. Significance is accepted when $P<0.05$. # P < 0.05 vs. control, * P < 0.05 significantly different from the folic acid-only treated group.

## Evaluation of organ and body weights

Data in "Table 1" showed significant loss of kidney weight in both FA and FA+COE (5 and 10mg/Kg) treated groups when compared to control on-treated group. However, groups receiving 20 mg/kg of COE either as treatment or prevention showed higher average kidney weight when compared to the FA treated group. There was no significant change in average heart weight among all groups, except for the FA+COE 20mg/kg treated groups where a significant increase in average heart weight was documented when compared to the FA treated group. As for the average body weight, a reduction of body weight was observed at day 30 only in rats receiving FA as compared to control rats (Fig 6).

## Histopathological findings of renal and cardiac tissues

Histopathological examination of the control group showed no specific changes or damage in kidney tissues (Fig 7A), while 50% of rats that received FA demonstrated mild interstitial chronic inflammation (Fig 7B). Pathology of rats receiving 5mg/kg of cannabis oil extract was overall unhanged when compared to the FA treated group, but with minor detectable interstitial chronic inflammation (Fig 7C). Interestingly, cannabis oil extract treatment at 10 and

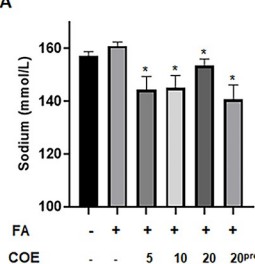
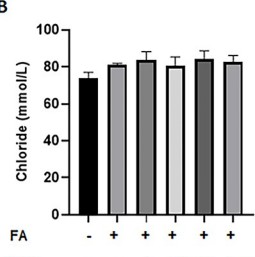
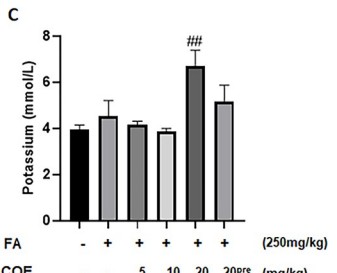

**Fig 5. Serum electrolyte levels of control and rats treated with folic acid and COE.** Rats were injected with 250 mg/Kg intraperitoneal folic acid on day 6. COE was administered eight days after the FA injection daily for 8 days then every other day for one week thereafter. In some experiments (COE 20^pre) rats were first injected with 20 mg/kg cannabis oil and at day 6 folic acid was administered. Sodium (A), Chloride (B) and Potassium (C) levels were measured. Each column represents the mean ± SEM of seven animals. Statistical analysis was performed using unpaired t-test. Significance is accepted when $P<0.05$. #$P<0.05$ vs. control, *$P<0.05$ significantly different from the folic acid-only treated group.

**Table 1. Average kidney and heart weight of rats.**

|  | Group 1 (Normal control, n = 5) | Group 2 (FA control, n = 5) | Group 3 (FA + 5mg/kg COE, n = 7) | Group 4 (FA + 10mg/kg COE, n = 6) | Group 5 (FA + 20mg/kg COE, n = 7) | Group 6 (20mg/kg COE + FA, n = 7) |
|---|---|---|---|---|---|---|
| Kidney weight (g) | 0.98±0.03 | 0.81±0.05 * | 0.82±0.04** | 0.85±0.03* | 1.04±0.03** | 0.95±0.04* |
| Heart weight (g) | 0.86±0.05 | 0.74±0.04 | 0.79±0.04 | 0.82±0.05 | 0.86±0.03* | 0.80±0.04 |

\* $P<0.05$. \*\* $P<0.01$ vs control group, \* $P<0.05$, \*\* $P<0.01$ vs FA group

20mg/kg showed a remarkable improvement in kidney structure whereby no significant findings of glomerular damage was observed in 75% of the analyzed samples. Focal tubular distention (Fig 7D) and mild vascular congestion (Fig 7E) was observed in 25% in the kidneys of rats treated with COE 5mg/kg and 10mg/kg respectively. Finally, 50% of the rat group treated with 20mg/kg of cannabis oil extract as prevention, showed normal kidney structure while the other half showed mild focal interstitial chronic inflammation (Fig 7F).

Histopathological examination of the heart sections of control rats showed no signs of damage or injury to the cardiac tissues (Fig 8A), while focal interstitial chronic inflammation was observed in 75% in the cardiac tissues of the FA treated group (Fig 8B). 25% of cardiac tissues of the FA group and treated with 5mg/kg COE showed normal pathology, and the remaining 75% still show chronic inflammation (Fig 8C). Cannabis treated rats at doses 10 mg/kg (Fig 8D) and 20mg/kg (Fig 8E) showed normal cardiac pathology with no detectable inflammation. Finally, the dose of 20mg/kg of COE given as prevention showed remarkable pathological

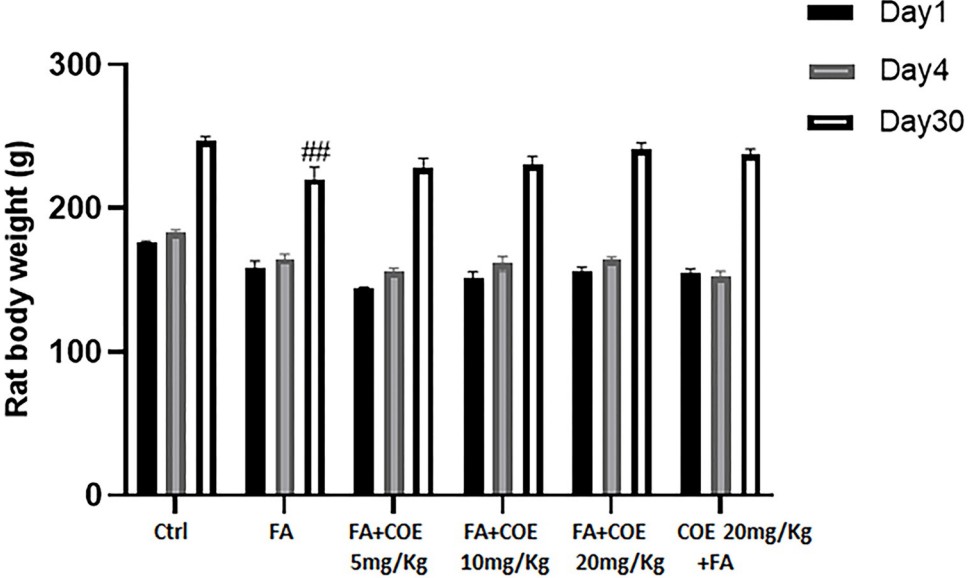

**Fig 6. Body weight of control and rats treated with folic acid and COE.** Rats were injected with 250 mg/Kg intraperitoneal folic acid on day 6. COE was administered eight days after the FA injection daily for 8 days then every other day for one week thereafter. In some experiments (COE 20mg/kg+FA) rats were first injected with 20 mg/kg cannabis oil and at day 6 folic acid was administered. Rats were weighted at day 1, 4 and 30. Each column represents the mean ± SEM of seven animals. Statistical analysis was performed using two way ANOVA. Significance is accepted when $P<0.05$. #$P<0.05$ vs. control at the same time point. Ctrl: control; FA: Folic acid; COE: Cannabis Oil extract.

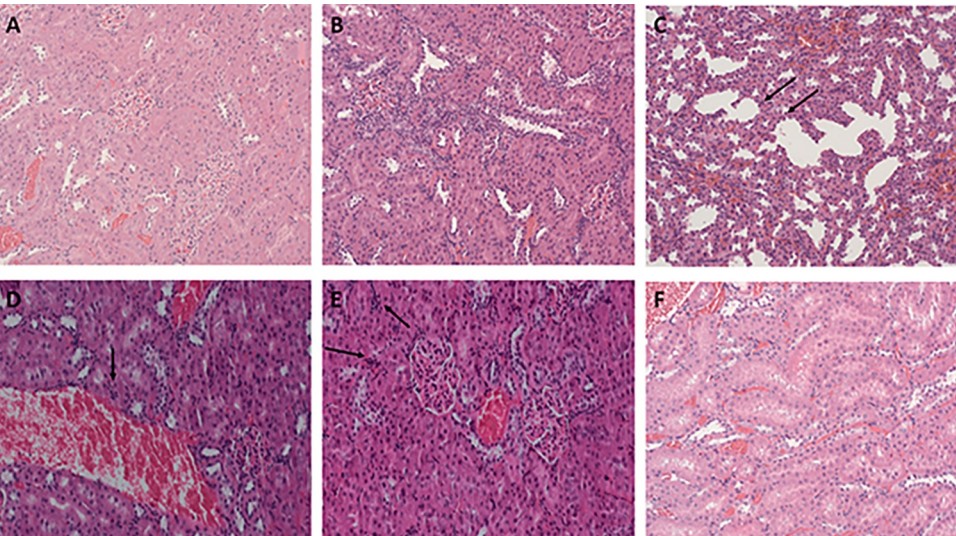

**Fig 7. Histopathology of renal sections of all rat groups (H&E stained).** (A) Control rats showing renal parenchyma with no specific changes (enlargement: ×200), in (B) Folic acid treated rats demonstrating mild interstitial chronic inflammation (enlargement: ×200). Cannabis treated rats at different doses (C) (5 mg/kg; enlargement: ×200), (D) (10 mg/kg; enlargement: ×200) and (E) (20 mg/kg; enlargement: ×200) showing mild interstitial chronic inflammation (C), Focal tubular distention (D) and mild vascular congestion (E) respectively. (F) Rats that have received a dose of 20mg/kg of cannabis oil extract as prevention (enlargement: ×200), showed mild focal interstitial chronic inflammation.

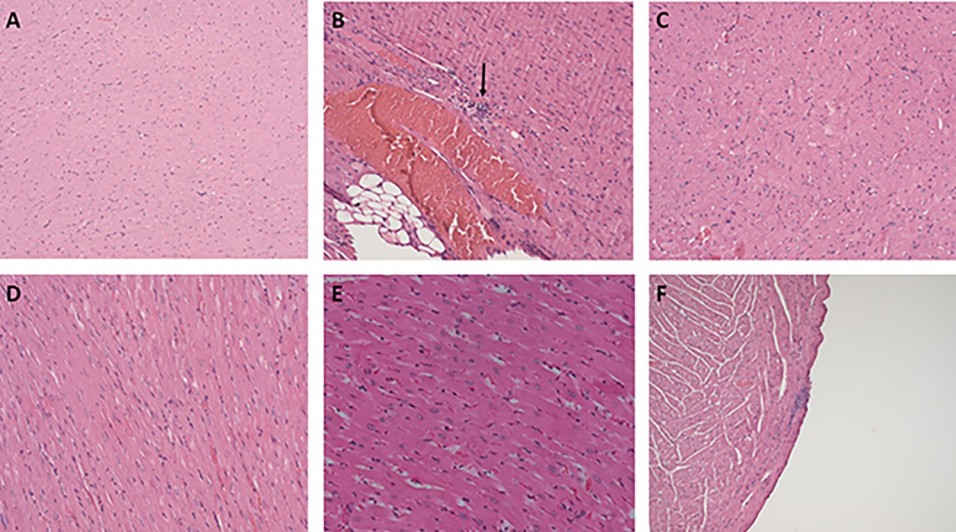

**Fig 8. Histopathology of heart sections of all rat groups (H&E stained).** (A)Control rats showing myocardium with no specific findings (enlargement: ×200) while (B) Folic acid treated rats demonstrating mild focal interstitial chronic inflammation (enlargement: ×200). Cannabis treated rats at different doses (C) (5 mg/kg; enlargement: ×200), (D) (10 mg/kg; enlargement: ×200) and (E) (20 mg/kg; enlargement: ×200) showing myocardium with no specific changes. (F) Rats that have received a dose of 20mg/kg of cannabis oil extract as prevention (enlargement: ×200), showed Mild focal interstitial chronic inflammation.

improvement whereby 75% of cardiac tissues presented normal cardiac pathology and only 25% showed mild focal interstitial chronic inflammation (Fig 8F).

## Discussion

Folic acid is a B vitamin consumed in the form of fortified foods (egg yolk, leafy vegetables and animal livers) or supplements plays an indispensable role in the synthesis of RNA and DNA molecules. At lower doses and under physiological conditions folate is freely filtered by the glomerulus and nearly 100% reabsorbed by folate receptor (folate receptor 1) abundantly expressed on the luminal side of proximal tubular epithelial cells. At higher doses, FA accumulate in larger amounts in the kidney inducing the onset of inflammation, necrosis, cell death, oxidative stress and fibrogenic changes in kidney epithelial cells. Accordingly, FA induced kidney fibrosis in rodents is a widely used in vivo model to study acute kidney injury and kidney fibrosis [20, 24].

To date, there is no effective treatment that could prevent or reverse renal fibrosis [19], stressing the urgent need for testing the efficacy of a variety of pharmacological agents capable of restoring the redox balance and possessing anti-inflammatory properties that can be beneficial in the management of renal fibrosis [25].

The important findings of our study suggested that acute exposure of podocytes to high concentrations of folic is associated with reduced cell viability and migration. Mechanistically, folic acid inhibited the phosphorylation of the prosurvival pathway AKT while inducing the phosphorylation of P38MAPK. Furthermore, this study confirmed the adverse effects of high concentrations of folic acid but also highlighted the protective effect of cannabis oil extract against Folic acid-induced acute cytotoxicity, impairment of cell migration and modulation of the underlying molecular pathways (AKT and P38 MAPK). Moreover, the in vitro results were supported in the *in vivo* model in which cannabis oil extract decreased serum creatinine and sodium levels induced by FA, as well as ameliorated renal and cardiac pathology.

The majority of the reports investigating FA induced renal fibrosis and AKI are using a well-established FA-rodent model, but to the best of our knowledge only two reports investigated the effect of high dose of FA using in vitro renal epithelial cell culture model [26, 27]. Since podocytes play a key role in the formation and maintenance of a functional glomerular filtration barrier, then we investigated the effect of high dose FA on podocytes biology.

Our data revealed that acute exposure (24 h) to higher concentrations of FA (range 1-25mM) was cytotoxic to rat podocytes in a dose dependent manner with an IC50 of 25mM. These results are in agreement with the only two studies reported investigating FA induced cytotoxicity in vitro. Our results corroborate the findings of the first study in which the cytotoxicity of FA was observed upon FA treatment of human proximal tubule cells (HK2) at 18 mM and 23 mM after 24h [27]. However, in the second study Kandel et al., (2022) used two human kidney cell lines (HK-2 and Caki-1 cells) and one rat cell line (NRK cells). Exposure of these cells to FA for 72h significantly inhibited the growth of both HK-2 and NRK cells by 20% and 22%, respectively [26]. Although acute exposure to higher concentrations of FA was cytotoxic to all kidney epithelial cell lines used, The difference in response between these three cell lines in one hand and our cell line in the other hand could be attributed to the time of exposure (24h vs 72hrs) and to the type of kidney cell line used (human vs rat, normal vs immortalized).

The biologic effect of COE depends on its chemical constituents which largely varies according to the country of origin and the geographic location from which it was harvested. Consistently, CBD, which is the major compound in COE, was shown to exert protective effect at lower concentration against different cytotoxicity models including cisplatin and sodium orthovandate (submitted work), hydrogen peroxide in hippocampal neuron culture [28], high

glucose-induced arrhythmia and cytotoxicity [29]. Also, it was found to alleviate UVB-induced cytotoxicity in human keratinocytes HaCaT cells [30]. Interestingly, in vitro analysis showed that COE protected podocyte cells against the cytotoxic effect of FA in a concentration-dependent manner.

As previously mentioned, administering a high dose of FA can quickly accumulate and form crystals in the renal tubules causing renal cortical scarring culminating in renal injury and subsequently decreased glomerular filtration rates (GFRs). While podocyte cell constitute a key cell in the formation and the maintenance of a functional glomerular filtration barrier, we went to further evaluate the effect of folic acid on podocyte cell migration. Importantly, our results demonstrated that FA attenuated podocyte cell migration in vitro. Worthy of note that the concentration used in wound healing assay for FA is not toxic to podocytes indicating that migration inhibitory the effect of FA is not related to the reduction in cell viability. In addition, COE was able to restore the migratory capacity in FA treated podocytes without altering cell viability. The latter protective effect of COE was previously reported in cisplatin and orthovandate treated podocytes (submitted work) and in several reports in which low concentrations of CBD significantly improved wound healing in non-cancerous cells including Human Brain Endothelial Cells and fibroblasts [31, 32].

In order to uncover the potential molecular mechanisms by which FA exerted its cytotoxic effect, we evaluated the phosphorylation of AKT and P38 MAPK. Of note, AKT is a podocyte pro-survival pathway and its inhibition by various stressor e.g., TGF-beta induce podocytes apoptosis. Furthermore, P38 MAPK is well-known pro-apoptotic pathway in kidney podocytes and its activation has deleterious effect on podocytes health [33]. Our findings corroborated the cell viability assay in which FA inhibited the phosphorylation of the pro-survival AKT while induced the phosphorylation of the pro-apoptotic p38 MAPK. These results are in agreement with previous studies demonstrating the same phosphorylation profile in a model of FA induced kidney injury [34] and in another model of transition of acute kidney injury into chronic injury [35].

In accordance to the cell viability and migration assays, the nephroprotective effect of COE on FA induced cytotoxicity is demonstrated by the induction of phospho-(S473)-AKT levels along with a significant reduction in phospho (T180 + Y182) P38 MAPK levels in a dose dependent manner. Of note, COE time course and dose-dependent induction of phospho-(S473)-AKT levels along with significant reduction in phospho (T180 + Y182) P38 levels was previously reported in another study in which the protective effect of CBD in an experimental multiple sclerosis model was mediated by increase in Phospho-AKT and a reduction in P38 MAPK activity [36].

We previously showed in *in vitro* and *in vivo* studies COE possess potent anti-inflammatory effects [17]. Cannabinoids exert their effect through the interaction with the receptors of the endocannabinoid system in the human body [37]. Both CB1 and CB2 receptors are expressed in the kidneys [38]. For instance, it was shown that CBD treatment had protective benefits against inflammatory and oxidative damage in renal ischemia/reperfusion models [39]. Moreover, besides THC and CBD, cannabis oil extract contains different components such as β-myrcene, cannabinol (CBN), and limonene that possess anti-inflammatory properties [17]. Myrcene which is present at a proportion of 1.94% in cannabis oil extract was shown to improve kidney function through the downregulation of oxidative stress and inflammation [17, 40]. Thus, we believe that the above mentioned beneficial effect of COE can be mediated in part through cannabinoid receptors. However, the exact contribution of these cannabinoid receptors in kidney biology requires further investigation in kidney disease models.

In keeping with kidney function, a rise in serum urea, creatinine and/or electrolytes can be indicative of a decrease in renal clearance [41]. Studies have shown that following FA

administration in mice, increased serum creatinine and urea [42]. In our current study, FA administration only increased serum creatinine while serum urea levels remained unchanged. Treatment with COE at a doses 5 and 10mg/kg remarkably reduced serum creatinine. On the other hand, the 20mg/kg dose of COE given either as treatment or prevention did not significantly decrease serum creatinine. These findings indicate that COE at doses of 5 and 10mg/kg could be a potential agent used to reverse the elevation of serum creatinine in renal disease. In fact, a study exploring the nephroprotective effect of CBD showed that treatment with 5 and 10mg/kg decreased serum creatinine and urea significantly in cisplatin induced nephrotoxicity mice model [43]. In addition, exposure to high doses of marijuana (25 and 50mg/kg) in which THC was the major constituent decreased plasma sodium levels in Wistar rats [44]. In our study, all doses of cannabis oil extract led to a significant decrease in serum sodium levels.

Our data showed that animals receiving FA alone had lower body weight at the end of the experiment when compared to control non-treated group. In particular FA-treated rats had lower kidney weight than the non-treated group while rats receiving 20 mg/kg of COE as treatment and prevention showed normal kidney weights. Gupta et al showed that the administration of FA at lower doses (100 mg/kg) to induce acute kidney injury caused significant weight loss in mice but a single administration of 250mg/kg of FA did not affect mice weight. It was also observed that renal hypertrophy and increase in kidney weight occurred following FA administration as a compensatory mechanism [45].

The beneficial effect of cannabis oil extract on renal morphology and function is not well understood. Histological examination of kidney sections showed interstitial inflammation, tubular distention and vascular dilatation in rats kidneys receiving either FA alone or FA followed by COE at a dose of 5mg/kg suggesting the presence of renal injury. These findings are in agreement with another study where a single dose of 250mg/kg of FA in rats had a toxic effect on the kidneys such as widening of Bowman's urinary space and dilatation of the tubules indicating the occurrence of tubular necrosis [46]. Also, inflammation activates of wound healing processes but sustained inflammatory stimuli can cause abnormal wound healing and scarring manifested as fibrosis [47]. Moreover, our results showed that treatment with higher doses of COE (10 and 20mg/kg) caused improvement in renal pathological findings. Pretreatment of animal with COE (preventive treatment) at a dose of 20mg/kg was able to completely resolve interstitial inflammation. Tubular distention and vascular dilatation were present in half of the sections while the other half showed complete recovery. Hence, these findings suggested that treatment with COE at a dose of 10 and 20mg/kg can be a promising intervention to reverse kidney damage induced by FA. Inflammation resolution in the groups receiving cannabis oil extract could be mediated by CBD, β-myrcene, cannabinol (CBN), and limonene that possess potent anti-inflammatory properties [17].

The modulation of cannabinoid receptors, more specifically CB1 and CB2 receptors play an important role in renal disease models [18]. In fact, CBD is considered as a CB1 antagonist and as a negative allosteric modulator of CB2 receptors. On the other hand, THC acts as a partial agonist at the CB1 and CB2 receptors [48]. Previous studies have shown that the activation and upregulation of CB1 receptors contributed to the development and worsening of kidney diseases [49]. Moreover, other studies have shown that blocking CB1 receptors or activating CB2 receptors protected against tubular damage by decreasing renal inflammation and oxidative stress [50, 51]. We believe, that antagonizing CB1 receptors and activating of CB2 receptors could explain the mechanism of action by which Lebanese cannabis oil extract caused improvement of kidney function and structure.

Finally, cardiorenal syndrome (CRS) is defined as disorders of the heart and kidneys wherein acute or chronic dysfunction in one organ may induce acute or chronic dysfunction of the other. CRS type 4 denotes cardiovascular damage in CKD patients at any stage. In CKD,

volume overload and pressure cause left ventricular hypertrophy that is accompanied by histological alterations and fibrosis. The structural modifications further lead to diastolic dysfunction and oxygen demand is increased [47]. Previous study showed alterations in cardiac function and structure following FA induced acute kidney injury [52]. However, few studies have explored the effects of CKD and renal fibrosis induced by nephrotoxic agents on the heart. Accordingly, we investigated the relationship between renal damage and cardiac damage in our *in vivo* model. Our data revealed that high doses of FA either given alone to induce renal fibrosis or followed by 5mg/kg of COE led to structural cardiac damage manifested as chronic inflammation. Alternatively, other showed that FA only causes injury to the kidneys without affecting other body organs [53]. The current study confirmed the presence of cardiorenal syndrome type 4. The administration of 10mg/kg of COE improved cardiac structures in all the treated samples. These findings are of considerable importance. However, further studies are needed in order to better understand Lebanese COE effect on cardiac physiology.

## Conclusion

The current findings demonstrated that COE was able to significantly reduced serum creatinine in rat model of FA—induced nephrotoxicity. The beneficial effect COE was also evident in kidney and cardiac pathology examination. Furthermore, COE showed promise as a preventive and therapeutic treatment against FA-induced nephrotoxicity. However, future studies are needed to elucidate the detailed molecular mechanisms underlying the kidney protective effects of Lebanese cannabis oil extract.

## Supporting information

**S1 Raw images.**
(PDF)

## Author Contributions

**Conceptualization:** Wissam H. Faour, Mohamad Mroueh.

**Formal analysis:** Diana Bylan, Alia Khalil, Selim Nasser.

**Funding acquisition:** Wissam H. Faour, Mohamad Mroueh.

**Investigation:** Diana Bylan, Alia Khalil, Wassim Shebaby, Christabel Habchy, Selim Nasser, Wissam H. Faour, Mohamad Mroueh.

**Methodology:** Diana Bylan, Alia Khalil, Wassim Shebaby, Christabel Habchy, Selim Nasser, Wissam H. Faour, Mohamad Mroueh.

**Project administration:** Wissam H. Faour, Mohamad Mroueh.

**Resources:** Wissam H. Faour.

**Supervision:** Wassim Shebaby, Mohamad Mroueh.

**Writing – original draft:** Diana Bylan, Alia Khalil, Wassim Shebaby, Christabel Habchy, Selim Nasser, Wissam H. Faour, Mohamad Mroueh.

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
