## [Decision Letter · Decision Letter 0]

28 Jun 2024

PONE-D-24-19651Lebanese Cannabis Oil Extract protected against Folic Acid-Induced Kidney Fibrosis in RatsPLOS ONE

Dear Dr. Faour,

Thank you for submitting your manuscript to PLOS ONE. After careful consideration, we feel that it has merit but does not fully meet PLOS ONE’s publication criteria as it currently stands. Therefore, we invite you to submit a revised version of the manuscript that addresses the points raised during the review process.

We look forward to receiving your revised manuscript.

Kind regards,

Francesca Baratta, PharmD, PhD

Academic Editor

PLOS ONE

Journal Requirements:

2. To comply with PLOS ONE submissions requirements, in your Methods section, please provide additional information regarding the experiments involving animals and ensure you have included details on (a) methods of sacrifice, (b) methods of anesthesia and/or analgesia, and (c) efforts to alleviate suffering.

3. Please expand the acronym “LAUGSR-PIRF” (as indicated in your financial disclosure) so that it states the name of your funders in full.

LAUGSR-PIRF

5. We note that your Data Availability Statement is currently as follows: All relevant data are within the manuscript and its Supporting Information files

Reviewers' comments:

Reviewer's Responses to Questions

**Comments to the Author**

1. Is the manuscript technically sound, and do the data support the conclusions?

Reviewer #1: Yes

Reviewer #2: Yes

Reviewer #3: Yes

2. Has the statistical analysis been performed appropriately and rigorously? 

Reviewer #1: Yes

Reviewer #2: Yes

Reviewer #3: Yes

3. Have the authors made all data underlying the findings in their manuscript fully available?

Reviewer #1: Yes

Reviewer #2: Yes

Reviewer #3: Yes

4. Is the manuscript presented in an intelligible fashion and written in standard English?

Reviewer #1: Yes

Reviewer #2: Yes

Reviewer #3: Yes

5. Review Comments to the Author

**Reviewer #1: **The manuscript deals with the effect of Lebanese Cannabis Oil Extract on Folic Acid-Induced Kidney Fibrosis.

It is well written and the methodology is rigorous. Some modifications are needed in order to improve it:

- Introduction: The sentence before the prevalence of dialysis in Lebanon has to be deleted. Only the prevalence has to be cited without announcing it.

- Discussion: The discussion section has to begin with the relevant results of the study.

**Reviewer #2:** The current work evaluated the protective effect of Lebanese cannabis oil extract against folic acid-induced renal injury both in vitro and in vivo. The authors have concluded that Lebanese cannabis oil extract significantly reduced serum creatinine in a rat model of folic acid-induced nephrotoxicity. The beneficial effect of Lebanese cannabis oil extract was also apparent in kidney and cardiac pathology examinations. Experimental design and statistical analysis are

Following are a few queries and suggestions.

1. Authors should support the results with good-quality photographs of cell culture showing wound healing.

2. The histological photomicrographs have been compared within the groups at different magnifications. Comparisons should be made at the same magnifications. The labelings on them are not clear.

3. Western blotting results should have been supported with immunofluorescence staining.

**Reviewer #3:** Adverse effects of Cannabis, should be stated in the introduction section.

Figures 2, 7 & 8 are not clear (blurred, and dark). Please consider updating.

Discussion: Results were occasionally repeated without proper explanation or discussion.

6. PLOS authors have the option to publish the peer review history of their article (what does this mean?). If published, this will include your full peer review and any attached files.

Reviewer #1: **Yes: **Mona MLIKA

Reviewer #2: **Yes: **Devendra Pathak

Reviewer #3: No

---

## [Author Response · Author response to Decision Letter 0]

2 Sep 2024

Dear Editor-in-Chief, PLoS One

We would like to thank you and the reviewers for their efforts in reviewing our manuscript and providing constructive comments. We answered each of the reviewers’ comments as described below and we made the necessary changes in the manuscript when applicable that can be found highlighted in red. Furthermore, we critically reviewed the paper for potential grammatical errors and made the necessary corrections

Reviewers' comments:

Reviewer's Responses to Questions

Comments to the Author

1. Is the manuscript technically sound, and do the data support the conclusions?

Reviewer #1: Yes

Reviewer #2: Yes

Reviewer #3: Yes

Answer: We thank the reviewers for their comments

2. Has the statistical analysis been performed appropriately and rigorously? 

Reviewer #1: Yes

Reviewer #2: Yes

Reviewer #3: Yes

Answer: We thank the reviewers for their comments

3. Have the authors made all data underlying the findings in their manuscript fully available?

Reviewer #1: Yes

Reviewer #2: Yes

Reviewer #3: Yes

Answer: We thank the reviewers for their comments

4. Is the manuscript presented in an intelligible fashion and written in standard English?

Reviewer #1: Yes

Reviewer #2: Yes

Reviewer #3: Yes

Answer: We thank the reviewers for their comments

5. Review Comments to the Author

Reviewer #1: The manuscript deals with the effect of Lebanese Cannabis Oil Extract on Folic Acid-Induced Kidney Fibrosis.

It is well written and the methodology is rigorous. Some modifications are needed in order to improve it:

- Introduction: The sentence before the prevalence of dialysis in Lebanon has to be deleted. Only the prevalence has to be cited without announcing it.

- Discussion: The discussion section has to begin with the relevant results of the study.

Answer to Reviewer 1: we thank the reviewer for this comments. accorinlgy, we made the necessary changes as requested

Reviewer #2: The current work evaluated the protective effect of Lebanese cannabis oil extract against folic acid-induced renal injury both in vitro and in vivo. The authors have concluded that Lebanese cannabis oil extract significantly reduced serum creatinine in a rat model of folic acid-induced nephrotoxicity. The beneficial effect of Lebanese cannabis oil extract was also apparent in kidney and cardiac pathology examinations. Experimental design and statistical analysis are

Following are a few queries and suggestions.

1. Authors should support the results with good-quality photographs of cell culture showing wound healing.

2. The histological photomicrographs have been compared within the groups at different magnifications. Comparisons should be made at the same magnifications. The labelings on them are not clear.

3. Western blotting results should have been supported with immunofluorescence staining.

Answer to Reviewer 2:

We would to thank the reviewer for the comments:

1. We uploaded better quality photographs of cell culture of the wound healing

2. We changed the photos so that all photo show the same magnifications. Also, we adjusted the labeling to make it clearer. We also, made the necessary in the figures legends 

3. We thank the reviewer for this comments. We used western blot to evaluate the phosphorylated activated levels of AKT and P38MAPK which is the most relevant in term of affecting cell survival. Although, immunostaining is important however, it will not provide a quantitative measurement of the activated kinase. It simply shows the localization of the kinase rather than its activated pattern. Furthermore, immunostaining of phosphorylated kinases is very challenging in particular when using paraffin embedded sections. Deparaffinization of paraffin section with xylene (a very strong organic solvent used to dissolve paraffin) prior staining with the relevant anti-phospho-protein antibody can cause a loss of the phospho group which bias the results. As such we preferred to do only western blot

Reviewer #3: Adverse effects of Cannabis, should be stated in the introduction section.

Figures 2, 7 & 8 are not clear (blurred, and dark). Please consider updating.

Discussion: Results were occasionally repeated without proper explanation or discussion.

Answers to Reviewer 3

We would to thank the reviewer for the comments. we uploaded better quality Figures 2, 7 & 8

We reviewed the discussion and made the necessary changes when applicable in addition to delete redundancies in the text

Respectfully,

Wissam, on behalf of all authors

---

## [Decision Letter · Decision Letter 1]

25 Sep 2024

Lebanese Cannabis Oil Extract protected against Folic Acid-Induced Kidney Fibrosis in Rats

PONE-D-24-19651R1

Dear Dr. Faour,

We’re pleased to inform you that your manuscript has been judged scientifically suitable for publication and will be formally accepted for publication once it meets all outstanding technical requirements.

Kind regards,

Francesca Baratta, PharmD, PhD

Academic Editor

PLOS ONE

Reviewers' comments:

Reviewer's Responses to Questions

**Comments to the Author**

1. If the authors have adequately addressed your comments raised in a previous round of review and you feel that this manuscript is now acceptable for publication, you may indicate that here to bypass the “Comments to the Author” section, enter your conflict of interest statement in the “Confidential to Editor” section, and submit your "Accept" recommendation.

Reviewer #1: All comments have been addressed

Reviewer #2: All comments have been addressed

2. Is the manuscript technically sound, and do the data support the conclusions?

Reviewer #1: Yes

Reviewer #2: Yes

3. Has the statistical analysis been performed appropriately and rigorously? 

Reviewer #1: Yes

Reviewer #2: Yes

4. Have the authors made all data underlying the findings in their manuscript fully available?

Reviewer #1: Yes

Reviewer #2: Yes

5. Is the manuscript presented in an intelligible fashion and written in standard English?

Reviewer #1: Yes

Reviewer #2: Yes

6. Review Comments to the Author

Reviewer #1: (No Response)

Reviewer #2: The current work evaluated the protective effect of Lebanese cannabis oil extract against folic acid-induced renal injury both in vitro and in vivo. The authors have made the desired changes to improve the manuscript. The authors provided good-quality photographs of cell cultures showing wound healing. The histological photomicrographs have been compared within the groups at the same magnifications. The reasons behind not doing immunofluorescence staining are reasonable. Given the quality of the revised manuscript, I recommend its acceptance for publication.

7. PLOS authors have the option to publish the peer review history of their article (what does this mean?). If published, this will include your full peer review and any attached files.

Reviewer #1: **Yes: **MONA MLIKA

Reviewer #2: **Yes: **Devendra Pathak

---

## [Editor Report · Acceptance letter]

2 Oct 2024

PONE-D-24-19651R1 

PLOS ONE

Dear Dr. Faour, 

I'm pleased to inform you that your manuscript has been deemed suitable for publication in PLOS ONE. Congratulations! Your manuscript is now being handed over to our production team.

Kind regards, 

on behalf of

Dr. Francesca Baratta 

Academic Editor

PLOS ONE